# Pathological Findings of Donor Vessels in Bypass Surgery

**DOI:** 10.3390/jcm13072125

**Published:** 2024-04-06

**Authors:** Yohei Nounaka, Yasuo Murai, Asami Kubota, Atsushi Tsukiyama, Fumihiro Matano, Kenta Koketsu, Akio Morita

**Affiliations:** 1Department of Neurological Surgery, Nippon Medical School Hospital, Tokyo 113-8603, Japan; ymurai@nms.ac.jp (Y.M.); ak0813@nms.ac.jp (A.K.); s00-078@nms.ac.jp (F.M.); amor-tky@nms.ac.jp (A.M.); 2Department of Neurological Surgery, Nippon Medical School Musashikosugi Hospital, Kawasaki 211-8533, Japan; atsuki-19@nms.ac.jp; 3Department of Neurological Surgery, Nippon Medical School Chiba Hokusou Hospital, Inzai 270-1694, Japan; kenta7240031@nms.ac.jp

**Keywords:** bypass surgery, superficial temporal artery, radial artery, moyamoya disease, pathology, dissection, atherosclerosis, STA-MCA bypass, bipolar, aneurysm, donor vessel

## Abstract

**(1) Background** Cerebral revascularization is necessary to treat intracranial arterial stenosis caused by moyamoya disease, atherosclerosis, or large complex aneurysms. Although various donor vascular harvesting methods have been reported safe, there are no reports on the histological evaluation of donor vessels for each disease, despite the variety of diseases wherein vascular anastomosis is required. **(2) Methods** Pathological findings of the superficial temporal artery (STA), radial artery (RA), occipital artery (OA), and saphenous vein (SV) harvested at the institution were analyzed. Patients classified according to aneurysm, atherosclerosis, and moyamoya disease were assessed for pathological abnormalities, medical history, age, sex, smoking, and postoperative anastomosis patency. **(3) Results** There were 38 cases of atherosclerosis, 15 cases of moyamoya disease, and 30 cases of aneurysm in 98 donor vessels (mean age 57.2) taken after 2006. Of the 84 STA, 11 RA, 2 OA, and 1 SV arteries that were harvested, 71.4% had atherosclerosis, 11.2% had dissection, and 10.2% had inflammation. There was no significant difference in the proportion of pathological findings according to the disease. A history of hypertension is associated with atherosclerosis in donor vessels. **(4) Conclusions** This is the first study to histologically evaluate the pathological findings of donor vessels according to disease. The proportion of dissection findings indicative of vascular damage due to surgical manipulation was not statistically different between the different conditions.

## 1. Introduction

Cerebral revascularization, utilizing extracranial vessels for intracranial arterial stenosis caused by moyamoya disease or atherosclerosis, could be necessary to reduce the risk of stroke [1]. In addition, revascularization may be required to temporarily supplement intracranial blood flow or compensate for sacrificing the main artery in cases involving the treatment of a large aneurysm or the removal of a tumor that implicates a blood vessel [2]. The superficial temporal artery (STA), occipital artery (OA), and radial artery (RA) are used as donor arteries in cerebral revascularization surgery, and harvesting these vessels is an essential step in performing cerebral revascularization surgery [3,4].

Numerous methods for donor vessel harvesting have been documented so far, encompassing endoscopically assisted techniques and ultrasonic scalpel approaches [5,6]. A retrospective study further examined the pathology of the STA harvested using the bipolar cutting/dissection method, providing histological evidence affirming the safety of the approach [7]. In these donor vessel revascularization procedures, vessel injury occurs during harvesting, and complications due to ischemic events have been reported [8]. However, there are no studies on the background and frequency of vascular injury.

This rare histopathologic study of donor vascular dissection has not examined the characteristics of the patient’s disease. Lusis et al. reported that atherosclerosis is a progressive disease characterized by the accumulation of lipids and fibrous elements in the large arteries. Atherosclerosis is the primary cause of heart disease and stroke and it is the underlying cause of about 50% of all deaths in Westernized societies. The etiology of atherosclerosis is very complex and can be classified into genetic and environmental factors. Genetic factors include elevated LDL/VLDL, low HDL, hypertension, family history, gender, systemic inflammation, and metabolic syndrome, while environmental factors include high-fat diet, smoking, lack of exercise, and infectious agents. These factors are intricately related [9]. Intracranial atherosclerotic stenosis is one of the greatest risks of recurrent stroke compared with other causes of stroke. The mechanisms by which intracranial atherosclerotic stenosis can cause stroke include plaque rupture with thrombosis that can produce artery-to-artery embolism or occlusion of the artery; hemodynamic compromise due to stenotic plaques; branch occlusive disease due to the intima impingement of the ostium of a penetrating artery; or a combination of these mechanisms. Additionally, it is increasingly recognized as a risk factor for silent brain infarctions and dementia [10]. Cerebrovascular revascularization has the potential to reduce the risk of recurrent stroke in hemodynamically compromised patients [11]. Han et al. reported cases of progressive stenosis and/or occlusion of the intracranial internal carotid artery diagnosed as moyamoya disease with coexisting atherosclerosis [12]. In relation to histological findings and diseases, there have been previous reports of intimal thickening in the STA with moyamoya disease [13], complications arising from abnormal STA grafts that could not be used as donor arteries for cerebral revascularization in middle cerebral artery stenosis [14,15], and damage to dissected vessels during the harvesting of donor vessels, resulting in anastomotic occlusion and intraoperative thrombosis [8]. There are limited reports on the effects of the histological findings of donor vessels on anastomosis, and there are no reports on the effects of underlying diseases, such as atherosclerosis, moyamoya disease, and aneurysms. Recognizing the importance of assessing the safety and risk associated with donor vessel harvesting for each specific disease, we sought to conduct a comprehensive evaluation by comparing the histological findings of each disease and the corresponding donor vessels. Our study focused on investigating the frequency of abnormal pathological findings in vessels harvested as donors for cerebral revascularization procedures and their subsequent effect on patency post-revascularization. If variations in the rate of donor vessel damage and its impact on patency are observed across different diseases, it may become imperative to tailor the harvesting method on a case-by-case basis.

The RA is used in neurosurgery for high-flow bypass procedures involving complex aneurysms situated between the middle cerebral artery and external carotid artery [3]. Notably, occlusion or thrombus formation within the graft can lead to severe complications [16]. Additionally, RA is commonly used in coronary artery bypass grafting, with some histological studies reported [17]. Yuan et al. conducted a comparative analysis of the pathology among the internal mammary artery, radial artery, and saphenous vein used as bypass grafts in coronary artery disease [18]. However, no studies have examined the histology of the STA, a typical donor vessel used in neurosurgery, on a disease-specific basis. Furthermore, in our study, we also histologically investigated the differences in the effects of the vascular harvesting maneuvers among the different vessels.

## 2. Materials and Methods

### 2.1. Surgical Procedure

Local anesthetics were deliberately avoided on the scalp due to concerns about potential damage to the donor vessel wall. The distal end of the donor vessel was identified using a handheld Doppler monitor and harvested under a microscope using the cut-down technique. To create spacers, scrolled gauze was positioned on both sides of the intended incision line on the donor vessel, and four pairs of skin retraction hooks were placed over the spacers. A skin incision was then made directly above the distal side using a no. 15 blade, with careful attention paid to avoiding direct damage to the donor artery beneath the skin during incision.

The donor vessel was dissected using bipolar or mosquito forceps to expose the top surface of the vessel, thereby facilitating the visibility of the vessel’s branches. Subcutaneous tissue was sparsely lifted with a hook, aiding in delineating the boundary between the donor vessels and surrounding tissue. The surrounding connective tissue was dissected, and bipolar forceps were used for coagulated incisions, with care taken not to perform bipolar dissection away from the vessel to prevent branch retraction. Large branches around the donor artery were incised with scissors after coagulation, whereas other connective tissue and thin branches were incised using bipolar coagulation.

During the separation of the STA from the temporal fascia, a cottonoid was placed under the donor vessel and pulled upward to create space for easier dissection. The dissected vessel was covered with a papaverine-soaked cottonoid to prevent vasospasm. Before vascular anastomosis, the donor scalp artery was cut to an appropriate length, and a sample from the distal side of the vessel was submitted for pathological examination (Figure 1).

In cases of STA-MCA double bypass, the frontal branch of the STA was harvested from the proximal to the distal side, starting from the galea side after raising the skin flap. The harvesting procedure remained consistent with the method described above.

The dissecting method was maintained consistently across each disease. Following the incision of the vessel at the distal end, the vascular lumen was filled with heparinized saline and pressurized to maintain tautness, preventing kinking, thrombus formation, and vasospasm. Prior to anastomosis, a thorough check ensured that the donor vessel was not twisted, and any intravascular thrombus was drained.

There were no cases of intraoperative graft failure, and the surgery was completed after confirming the anastomosis for the target vessels.

### 2.2. Histological Study

We compared the histological findings, primary disease, history (hypertension, dyslipidemia, and diabetes), and smoking history of donor vessels obtained from 87 patients who underwent cerebral revascularization at our hospital from 2006 to 2022 retrospectively. The abnormal pathological findings included atherosclerosis, inflammatory infiltrates, and dissection. All cases were confirmed using hematoxylin and eosin staining, and skilled pathologists diagnosed all pathological findings at our institution. Cases with inflammatory cell infiltration in the wall were defined as “inflammation”, cases with fibrosis and thickening of the intima and tunica media as “atherosclerosis”, and cases with dissection in the wall as “dissection” (Figure 2).

The primary disease, pathological findings, medical history (hypertension, diabetes, and dyslipidemia), age, sex, smoking history, and postoperative patency were retrospectively investigated. The same surgeon performed all harvesting and anastomosis procedures, and postoperative patency was assessed using magnetic resonance imaging (MRI) or 3-dimensional computed tomography angiography (3DCTA) within 1 week and 1 year after bypass.

### 2.3. Statistical Analysis

A one-way analysis of variance was performed to determine the relationship between the disease and the patient’s age. The χ^2^ or Fisher’s exact test was performed for sex, pathology (inflammation, arteriosclerosis, and dissection), patency, and past medical history (hypertension, dyslipidemia, diabetes, and smoking). Other items not described in detail and that could not be classified were excluded from statistical analysis. Statistical analyses were performed using the IBM SPSS Statistics version 27. Statistical significance was set at *p* < 0.05 significance.

## 3. Results

A total of 98 donor vessels were collected since 2006. The specimens were classified by disease: 38 in the atherosclerosis group, 15 in the moyamoya group, 42 in the aneurysm group, and 3 in the other group. The other three cases were dural arteriovenous fistula, in which the STA was harvested as a safe bypass for treatment; internal trapping was performed after STA-MCA double bypass in a case of internal carotid artery injury during endonasal endoscopic trans-sphenoidal surgery, and transient ischemic attack due to the dissection of the middle cerebral artery in which STA-MCA anastomosis was performed. These three cases were determined not to belong to any of the diseases in the present study.

There were 38, 15, and 30 vessels in the arteriosclerosis, moyamoya, and aneurysm groups, respectively. The overall mean age was 57.2 years. The mean age for each disease group was 60.8 years for the atherosclerosis group, 34.1 years for the moyamoya group, and 61.1 years for the aneurysm group. Additionally, the patients in the moyamoya group were significantly younger than the other two groups (*p* < 0.001). The vessels harvested were the STA in 84 vessels, RA in 11 vessels, OA in 2 vessels, and the saphenous vein in 1 vessel. Forty-six patients were male, and the percentage of males in the atherosclerosis group was significantly higher than in the other disease groups.

Overall, abnormal pathological findings were observed in 70 (71.4%), 11 (11.2%), and 10 (10.2%) vessels with atherosclerosis, dissection, and inflammation, respectively. For each pathological finding, 4 (10.5%), 1 (6.7%), and 4 (9.5%) vessels of inflammation, 30 (78.9%), 11 (73.3%), and 28 (66.7%) vessels of arteriosclerosis, and 6 (15.8%), 1 (6.7%), and 3 (7.1%) vessels of dissection were observed in the atherosclerosis group, the moyamoya group, and the aneurysm group, respectively. There were no significant differences in inflammation, atherosclerosis, or dissection frequencies between the disease groups.

Regarding the patency of the anastomosis, the postoperative patency of the donor vessel was confirmed in 82 (96.5%) of the 85 patients. The anastomotic patency rate was 38 of 38 (100%) in the atherosclerosis group, 15 of 15 (100%) in the moyamoya group, and 27 of 30 (90%) in the aneurysm group, showing no significant difference; however, the anastomotic patency rate tended to be lower in the aneurysm group. Twelve vessels harvested in the aneurysm group were ultimately unused for anastomosis, as a precaution for potential revascularization, which proved unnecessary.

Examining underlying diseases, hypertension history was present in 56 (57.1%) cases, with a higher prevalence in the atherosclerosis group (29, 76.3%) compared to the moyamoya (5, 33.3%) and aneurysm (22, 52.4%) groups, showing statistical significance. Dyslipidemia was noted in 38 (38.8%) patients, with a significantly higher percentage in the atherosclerosis group (27, 71.1%) compared to the moyamoya (3, 20%) and aneurysm (8, 19.0%) groups. Fifteen cases (15.3%) had a history of dyslipidemia, twelve (31.6%) were in the atherosclerosis group, two (13.3%) were in the moyamoya group, and one (2.4%) was in the aneurysm group. The atherosclerosis group tended to be more common than the aneurysm group (*p* < 0.001). There were no significant differences in smoking rates (Table 1).

Statistical analyses were performed for the presence or absence of abnormal histological findings (inflammation, atherosclerosis, and dissection), age, sex, underlying disease, smoking history, and occlusion rate. Among the 10 patients with inflammation, the mean age was 55.5 years, with 6 (60%) being male. The vascular types included eight STA, one SV, and one OA. All bypasses were open, and no pre-existing diabetes was reported. Among these patients, six (60%) had hypertension and three (30%) had dyslipidemia. Seventy patients diagnosed with arteriosclerosis had a mean age of 58.1 years, with 36 (51.4%) being male. Of these, 61 had STA, 7 had RA, 1 had SV, and 1 had OA. Patency was confirmed in 60 cases, occluded in 2 cases, and 8 cases were not used for bypass. Pre-existing diabetes mellitus was present in 12 patients (17.1%), hypertension in 47 (67.1%), and dyslipidemia in 31 (44.3%). Eleven patients with dissection had a mean age of 61.9 years, with four (36.4%) being male, and all vessels were STA. Patency was confirmed in eight cases, occlusion occurred in one case, and two cases were not used for bypass. Among these patients, two (18.2%) had diabetes mellitus, eight (72.7%) had hypertension, and seven (63.6%) had dyslipidemia. No significant differences were observed between the presence of inflammation and dissection. However, hypertension was significantly more frequent in cases with arteriosclerosis (*p* = 0.002).

The STA and RA groups were compared in terms of mean age, sex, pathology (inflammation, atherosclerosis, and dissection), patency, and history (HT, DL, DM, and smoking history).

A total of 84 specimens were obtained in STA and 11 in RA.

There were significantly more women in the RA group; however, no other significant differences were observed. Although the pathological findings did not reach statistical significance, 61 (72.6%) patients with STA had atherosclerosis, 11 (13.1%) had dissection, and 8 (9.5%) had inflammation, whereas 7 (63.6%) patients with RA had atherosclerosis but no inflammation or dissection. The anastomotic patency rate was 69 of 72 (95.8%) in the STA and 11 of 11 (100%) in the RA. A total of 12 vessels in the STA were not used for bypass (Table 2).

## 4. Discussion

Since Yaşargil pioneered the STA-MCA anastomosis in 1969 [19], the STA has been a staple for cerebral revascularization in various diseases such as atherosclerosis, moyamoya disease, and aneurysms. Over the years, numerous reports have documented revascularization procedures using donor vessels, expanding beyond the STA. Noteworthy examples include the radial artery [20] and the saphenous vein [21], employed in treating complex intracranial aneurysms, and the occipital artery, utilized for addressing diseases in the posterior circulations [4]. Harvesting the donor vessel is a crucial procedure, as graft occlusion can lead to fatal complications [8].

Various methods for harvesting donor vessels have been reported in the previous literature [5,6,7]. Tokugawa et al. introduced the bipolar cutting method, utilizing bipolar instruments for dissecting the connective tissue and branches of the donor vessel. This method was praised for its simplicity, as the procedure involved dissecting, coagulating, and cutting tissue simultaneously using only bipolar instruments [7].

Wada et al. reported a harvesting and skeletonization technique for STA harvesting, using an ultrasonic scalpel originally designed for cardiac surgery. Dissection involved moving the ultrasonic scalpel along the vessel to expose the STA’s adventitia. Branches were obstructed and divided by ultrasonic protein coagulation without injuring the main trunk. Applied to 18 patients, this procedure demonstrated minimal bleeding and no thermal damage to the STA on histological observations, suggesting potential benefits in reducing operative time and morbidity [6].

Kubo et al. utilized an endoscopic method with a small incision for harvesting the STA. This technique required a 7 cm linear incision along the parietal branch. During dissection, the perforator of the frontal branch was exposed, coagulated with bipolar forceps, and severed with Metzenbaum scissors under endoscopic observation. The branch was then harvested subcutaneously. This approach allows for STA-MCA double anastomosis with a small incision, potentially avoiding postoperative scalp necrosis and hair loss [5].

This study represents the first examination of the pathology of intraoperative donor arteries specific to each underlying disease requiring cerebral revascularization surgery. While we initially considered the potential for variations in donor artery pathology findings, including vascular injury, based on the type of disease, our analysis revealed no significant differences. Hypertension and dyslipidemia emerged as significant factors strongly associated with atherosclerotic changes in the donor arteries which is similar to the previous report that hypertension and dyslipidemia are associated with atherosclerosis as genetic and environmental factors of atherosclerosis. Hurtubise et al. reported that hypertension and dyslipidemia are pathological conditions that damage the endothelium, triggering cell proliferation, vascular remodeling, apoptosis, and increased cellular permeability with increased adhesion molecules that bind monocytes and T lymphocytes to create a vicious cocktail of pathophysiological factors. These factors are induced in the vascular intima by chemoattractants and inflammatory cytokines, where monocytes differentiate into macrophages that take up oxidized LDL in an unregulated manner, forming foam cells and atherosclerotic lesions [22].

Diabetes mellitus has also been reported to be a factor in arteriosclerosis and was significantly higher in the arteriosclerosis group than in the aneurysm group, a result consistent with previous reports, but not significantly different from the moyamoya disease group [9].

In addition, there were cases of arteriosclerotic changes in the vessels of moyamoya disease, suggesting that arteriosclerosis may coexist with moyamoya disease. Because of the high risk of obtaining pathology from intracranial blood vessels, there have been reports of using high-resolution MRI to diagnose the possible coexistence of moyamoya disease and atherosclerosis; however, in cases in which revascularization procedures are performed, the investigation of pathology samples obtained intraoperatively may also be an additional aid in diagnosis [10].

The results of this study indicate that the frequency of vascular injury, particularly with the bipolar dissection method, remains consistent across underlying diseases. In contrast to a previous retrospective study where 38 specimens treated with bipolar cutting/dissecting methods showed no apparent dissection [7], our study identified dissections in 11 cases (11.2%) out of 98 specimens upon pathological examination. However, none of these cases resulted in bypass occlusion. These results underscore the necessity of acknowledging the inherent risk of dissection associated with bipolar cutting/dissecting methods during donor vessel harvesting. A prior report by Kimura et al. highlighted that the free edge of the intima and the exposed smooth muscle layer may induce thrombus formation, leading to acute bypass occlusion [8]. Therefore, meticulous attention to dissection during donor vessel harvesting and a thorough assessment of the vessel wall before anastomosis are imperative. Matsumura et al. suggested that suturing the dissected vessel wall with 11-0 nylon sutures is beneficial for repairing arterial dissection, even if observed after recutting the stump of the donor vessel [23].

Several reports highlighted the occurrence of iatrogenic pseudoaneurysms in different scenarios. Zheng et al. reported iatrogenic pseudoaneurysms of the STA resulting from vessel penetration with a needle or hook, or injury to the vessel wall during craniotomy [24]. Protecting vessels during harvesting and conducting thorough imaging evaluations post-bypass surgery are emphasized, as illustrated by a past case where pseudoaneurysm rupture after STA-MCA anastomosis caused massive intracerebral hemorrhage [25]. In addition, Srinivasan et al. reported pseudoaneurysms in the OA after posterior fossa craniotomy, underscoring the importance of vigilance regarding vessel wall damage regardless of the donor vessel type [26].

In the context of giant cell arteritis, Tian et al. reported inflammatory infiltrates in the STA vessel wall, consisting of lymphocytes, macrophages, and multinuclear giant cells, leading to almost occluded vessel lumens [27]. Despite considering the potential impact of inflammatory infiltrates on bypass occlusion, our study revealed no significant differences. However, given the limited number of cases with inflammatory cell infiltration in the vessel wall (10 out of 98), further investigations with a larger sample size and long-term patency rate evaluations are warranted for comprehensive understanding in the future.

Winkler et al. reported that bypass surgery for aneurysms has a significantly higher complication rate, including iatrogenic stroke, than bypass surgery for moyamoya disease or occlusive disease [28]. It is difficult to evaluate because some cases in the aneurysm group are used to temporarily supplement blood flow [29], and many cases in the aneurysm group, such as moyamoya disease or occlusive disease, are not treated with antithrombotic agents preoperatively; the possibility of obstruction due to decreased demand is also considered. This study did not separately explore the direction of harvesting between the parietal and frontal branches of the STA. Future investigations will consider a detailed comparison of the frontal and parietal branches individually.

No studies have directly compared the pathologies of the STA and RA. Although our study did not reach significance, we found dissection and inflammation only in the STA group; dissection with STA may be more prone to dissection due to mechanical damage. However, the small number of cases in the RA group requires additional study with more patients. Although the difference in patency was not significant, 69 of the 72 patients (95.8%) in the STA group showed patency, whereas all 11 patients in the RA group showed patency.

In a previous report, the mean diameter of the RA was documented as 2.6 mm [30], while that of the STA was noted to be 1 mm [31]. This discrepancy suggests that performing bypass surgery using the STA may require more attention to potential occlusion risks.

Although the number of OA and SV specimens obtained in this study was too small for meaningful comparison, the existing literature indicates that anastomosis in the posterior circulation is technically more challenging than that in the anterior circulation. Conducting further pathological studies with a larger number of cases may be imperative for comprehensive insights [32].

## 5. Limitation

It is noteworthy that our histological findings focused solely on the distal end of the donor vessels. While the entire length of the donor vessels was not examined, Tokugawa et al. reported that the distal part of the donor vessels is typically narrow and adherent to the surrounding tissue, making it a challenging area for pathological study [7]. Since the sample size in this study is still small, further increasing the sample size in the future will enhance the reliability of this study.

In this study, while bypass patency was confirmed in all moyamoya disease and atherosclerosis cases, the obtained patency results extended up to 1 year. However, there are reports of graft occlusion occurring more than 1 year later [32]. Investigating the relationship between disease, pathology, and long-term patency needs additional studies for a more thorough understanding.

## 6. Conclusions

This study marks the first attempt to examine the pathology specific to each underlying disease. Notably, no significant differences in pathological findings between diseases were observed; however, hypertension emerged as a strong correlate with atherosclerotic changes. Similarly, no significant differences were noted in the presence of inflammation, arteriosclerosis, and dissection nor in the patency of the bypass.

While the number of cases is limited, it is noteworthy that there were no instances of dissection or inflammation in the RA group.

No pathologic differences were observed in various pathologies including atherosclerosis, moyamoya disease, and aneurysms. These results suggest that careful dissection and anastomosis regardless of the underlying disease are important for a successful bypass procedure.

To comprehensively understand the relationship between the long-term prognosis and pathological findings of bypass patency, future evaluations should be conducted.

## Figures and Tables

**Figure 1 jcm-13-02125-f001:**
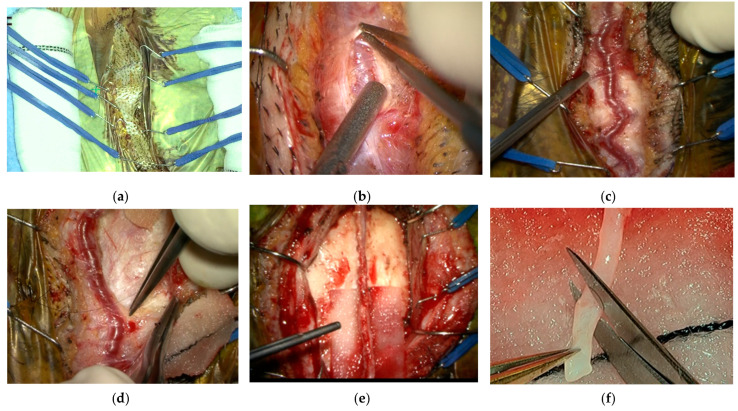
Surgical procedure for harvesting the donor vessel. (**a**) Scrolled gauze was placed on both sides of the incision line, and four pairs of retraction hooks were placed over the gauze. (**b**) Identification of the donor vessel and capturing its top surface. (**c**) Dissection of the top surface for enhanced visibility of the vessel’s branches. (**d**) Bipolar dissection of connective tissue and branches surrounding the vessel. (**e**) Harvesting the donor vessel. (**f**) The distal end of the donor vessel was cut away and used for pathological examination.

**Figure 2 jcm-13-02125-f002:**
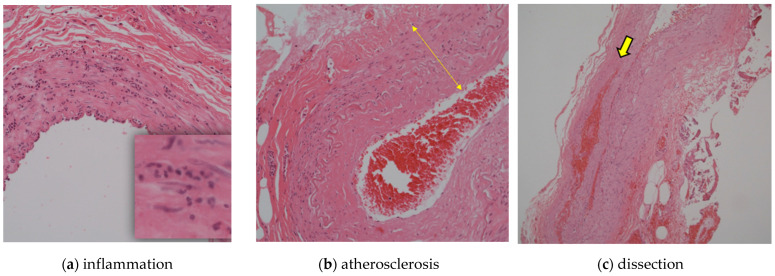
Histological study. (**a**) Inflammatory cell infiltration in the wall was defined as inflammation; (**b**) arrowhead shows fibrosis and thickening of the intima and tunica defined as atherosclerosis; (**c**) arrowhead shows dissection in the wall.

**Table 1 jcm-13-02125-t001:** Comparison between the atherosclerosis group, moyamoya disease group, and aneurysm group. There were significant differences between the colored groups.

	Atherosclerosis	Moyamoya	Aneurysm	Others	All
*n*	38	15	42	3	98
Average	60.8	34.1	61.1	72	57.2
SD	9.9	15.3	15.1	6.1	16.4
			<0.001		
		<0.001			
Vessel					
STA	38	15	28	3	84
RA	0	0	11	0	11
OA	0	0	2	0	2
SV	0	0	1	0	1
Sex					
Female	14	11	25	2	52
		0.017			
			0.043		
Pathology					
Inflammation	4	1	4	1	10
Arteriosclerosis	30	11	28	1	70
Dissection	6	1	3	1	11
Patency	38	15	27	2	80
Occlusion	0	0	3	0	5
HT	29	5	22	0	56
			0.026		
		0.003			
DL	27	3	8	0	38
			<0.001		
		<0.001			
DM	12	2	1	0	15
			<0.001		
Smoke	22	5	21	0	48

**Table 2 jcm-13-02125-t002:** The comparison of superficial temporal artery (STA) and radial artery (RA).

	STA	RA	*p*-Value
N	84	11	
Average	56.7	61.9	-
SD	16.5	16.9	
Sex (female)	41	9	*p* = 0.039
Pathology			
Inflammation	8	0	-
Arteriosclerosis	61	7	-
Dissection	11	0	-
Patency	69	11	-
Occlusion	3	0	
HT	48	6	-
DL	35	3	-
DM	15	0	-
Smoke	44	3	-

## Data Availability

Data is unavailable due to privacy or ethical restrictions.

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
