# Peer review of "Pathological Findings of Donor Vessels in Bypass Surgery"

_jcm, 2024, doi:10.3390/jcm13072125_

Round 1
Reviewer 1 Report (Previous Reviewer 2)
Comments and Suggestions for Authors
Thanks for the revisions and the current version has been improved significantly. Only one concern remains.
Please add a limitation part with potential future directions to help the readers get better understanding of this topic. Moreover, enlarging sample size to replicate the current findings should be strongly encoured.
PS. Please highlight the revised parts in the revised manuscript in the next round.
Comments on the Quality of English LanguageMinor editing of English language required
Author Response

Reviewer 2 Report (New Reviewer)
Comments and Suggestions for Authors
Authors present an analysis of pathological changes in 15 cases of moyamoya disease, and 30 cases of aneurysm in 98 donor vessels. Of superficial temporal artery (STA), radial artery (RA), occipital artery (OA), and saphenous vein (SV) 71.4% had atherosclerosis, 11.2% had dissection, and 10.2% had inflammation. Anastomotic patency rate tended to be lower in the aneurysm group The proportion of dissection findings indicative of vascular damage due to surgical manipulation was not statistically different between the different conditions. Is this a prospective study? According to our knowledge, it is not usual to take a specimen for pathohistological analysis of a donor vessel; is there an Ethics committee statement? How many graft failures did these patients have? Major drawback is examination of the distal end of the specimen; which is not representative - I suggest to include this in the title. Please include clinical relevance of this study and changes to the daily practice according to results; what are the future advances?
Comments on the Quality of English LanguageModerate changes.
Round 2
Reviewer 1 Report (Previous Reviewer 2)
Comments and Suggestions for Authors
Thanks for the revisions and no further concerns.
Comments on the Quality of English LanguageMinor editing of English language required
Reviewer 2 Report (New Reviewer)
Comments and Suggestions for Authors
Authors have sufficiently responded to reviewer remarks.
Comments on the Quality of English LanguageMinor editing.
This manuscript is a resubmission of an earlier submission. The following is a list of the peer review reports and author responses from that submission.
Round 1
Reviewer 1 Report
Comments and Suggestions for Authors
This paper needs significant improvement in terms of design and presentation! The goal of the paper is not clearly understandable and the results are presented in a very confuse way. The collected data seem to be very interesting but the scientific evaluation and analysis is very insufficient. The conclusion of the entire study should be clearly written.
Comments on the Quality of English LanguageThe paper contains many typos (already in the title!) and has a very poor quality of English. A significant improvement should be performed.
Reviewer 2 Report
Comments and Suggestions for Authors
“The pathological findings of donor vessels in extracraniai to intracranial bypass surgery”( jcm-2834085)
This manuscript aimed to histologically evaluate the pathological findings of donor vessels according to disease (atherosclerosis, Moyamoya disease and aneurysm). Pathological findings of the superficial temporal artery (STA), radial artery (RA), occipital artery (OA), and saphenous vein (SV) harvested at the institution were analyzed. The results revealed that no significant difference in the proportion of pathological findings according to the disease. A history of hypertension is associated with atherosclerosis in donor's vessels. Overall, this topic might be interesting. However, some concerns appeared after reading the whole manuscript.
1. “there are no reports of the histological evaluation of each disease.”this is not necessary the reason to conduct the current investigation. The importance of this topic either in medical practice or in theory should be provided to help the readers realize why research attention should be attracted to this topic. After reading the whole manuscript, it is still confusing me why you conduct this investigation. The current version of manuscript seems only focus a narrow topic which only can attract limited audience.
2. How did you determine the sample size? Did you calculate the sample size needed before formal study? The current sample size seems too little to get reliable results, especially the little sample size in RA group (N=11), which would seriously jeopardize the soundness of current investigation. The null results revealed in this manuscript might be probably due to the small sample size.
3. Please provide the effect size where available.
4. Please redesign table 1 and 2 to be more informative and exclude confusing. Especially table 1, I cannot catch up what the authors want to say in this table and what does the grey shade mean and why “<.001” not always with “*”, and what condition did you compare about each p value.
5. I strongly recommend that the paper be thoroughly proofread and edited for languages and grammars, to enhance readership, especially the logic of writing should be reorganized.
Comments on the Quality of English Language
English very difficult to understand/incomprehensible.